# Ultra-narrow room-temperature emission from single CsPbBr$_3$ perovskite quantum dots

Gabriele Rainò [1,2✉], Nuri Yazdani [3], Simon C. Boehme [1,2,4], Manuel Kober-Czerny [1,2], Chenglian Zhu[1,2], Franziska Krieg[1,2], Marta D. Rossell [5], Rolf Erni [5], Vanessa Wood [3], Ivan Infante[4,6] & Maksym V. Kovalenko [1,2✉]

Semiconductor quantum dots have long been considered artificial atoms, but despite the overarching analogies in the strong energy-level quantization and the single-photon emission capability, their emission spectrum is far broader than typical atomic emission lines. Here, by using ab-initio molecular dynamics for simulating exciton-surface-phonon interactions in structurally dynamic CsPbBr$_3$ quantum dots, followed by single quantum dot optical spectroscopy, we demonstrate that emission line-broadening in these quantum dots is primarily governed by the coupling of excitons to low-energy surface phonons. Mild adjustments of the surface chemical composition allow for attaining much smaller emission linewidths of 35−65 meV (vs. initial values of 70–120 meV), which are on par with the best values known for structurally rigid, colloidal II-VI quantum dots (20−60 meV). Ultra-narrow emission at room-temperature is desired for conventional light-emitting devices and paramount for emerging quantum light sources.

[1] Department of Chemistry and Applied Biosciences, Institute of Inorganic Chemistry, ETH Zurich, 8093 Zurich, Switzerland. [2] Laboratory for Thin Films and Photovoltaics, Empa−Swiss Federal Laboratories for Materials Science and Technology, CH-8600 Dübendorf, Switzerland. [3] Department of Information Technology and Electrical Engineering, ETH Zurich, Zurich 8092, Switzerland. [4] Department of Theoretical Chemistry, Faculty of Science, Vrije Universiteit Amsterdam, 1081 HV Amsterdam, The Netherlands. [5] Electron Microscopy Center, Empa–Swiss Federal Laboratories for Materials Science and Technology, CH-8600 Dübendorf, Switzerland. [6] Department of Nanochemistry, Istituto Italiano di Tecnologia, Via Morego 30, 16163 Genova, Italy. ✉email: rainog@ethz.ch; mvkovalenko@ethz.ch

Colloidally synthesized semiconductor quantum dots (QDs) are essential building blocks for diverse optoelectronic applications[1]. As a result of the tremendous effort over the last three decades, QDs can now be produced with nearly 100% photoluminescence quantum yields (PL QYs)[2,3], with narrow size distributions and facile emission tunability over the entire visible spectrum. For these reasons, colloidal QDs have become a light-emitter of choice for the latest generations of commercial LCD color displays[4] and are actively pursued for light-emitting diodes[5], lasers[6,7], and luminescent solar concentrators[8]. For display technologies, the narrow emission spectrum of QDs, i.e. with a linewidth (also referred to as full width at half maximum, FWHM) of <200 meV (ca. 40 nm in green for InP QDs) was a key enabler for entering and conquering market opportunities. This is also of vital importance in active pixel display technologies where OLED and QLED are the main pursued technological solutions.

A few years ago, a novel material system entered the scene—QDs of perovskite APbX₃ compounds [A = formamidinium (FA), methylammonium (MA), or Cs; X = Cl, Br, I]. They feature near-100% PL QY in solutions, and narrowband luminescence adjustable over the whole visible wavelength range[9–11]. High emission rates and large absorption coefficients place these QDs amongst the brightest known emitters. Their intrinsic defect-tolerance allowed for achieving these characteristics with utmost synthesis simplicity. Thus far, these QDs have been used in highly efficient LEDs approaching the theoretical external quantum efficiency (EQE > 20%)[12], as well as in solar cells[13], efficient lasers[14] and highly coherent sources of single[15,16] or bunched photons[17].

Emission lines of an isolated atom, not interacting with any matrix, as found, e.g. in atomic vapors, are extremely narrow, mainly limited by the radiative lifetime (Fourier-transform-limited linewidth, Fig. 1a)[18]. On the contrary, when the emissive state exists in a solid-state material, be it a localized atomic transition in a crystalline host or a delocalized exciton in a semiconductor, its emission linewidth is broadened by the coupling to the crystal vibrations (phonons, Fig. 1a). Further complexity in a single colloidal QD emerges from the strong quantum confinement, which splits the ground-state exciton band-edge manifold (often referred to as fine structure splitting, FSS; Fig. 1a). Depending on the energy differences among these states and the temperature of the system, emission can originate from multiple electronic transitions[19,20]. In addition, in an ensemble of QDs, structural disorder, e.g. size and/or composition inhomogeneity, can further broaden the emission line (inhomogeneous broadening, Fig. 1a). Structural defects in the core and surface regions of the QDs[21] or strain[22] are additional factors influencing the broadening of PL spectra. Disentangling the interplay of all these mechanisms in colloidal QDs has been a challenge of the last 30 years. Unveiling and reducing emission line-broadening would be greatly important for the development of ultra-narrow perovskite-based LEDs at the ensemble[23] and single-photon level, whereas ultra-narrow and spectrally tunable emission could allow single-photon multiplexing schemes[24], significantly boosting the transfer rates. Additionally, ultra-narrow emission could facilitate the co-integration of QD-based single-photon sources with atomic quantum memories[18].

Here, we perform single QD spectroscopy and abinitio molecular dynamics (AIMD) simulations for differently sized CsPbBr₃ QDs and demonstrate that emission line-broadening in these QDs is dominated by the coupling of the exciton to low-energy

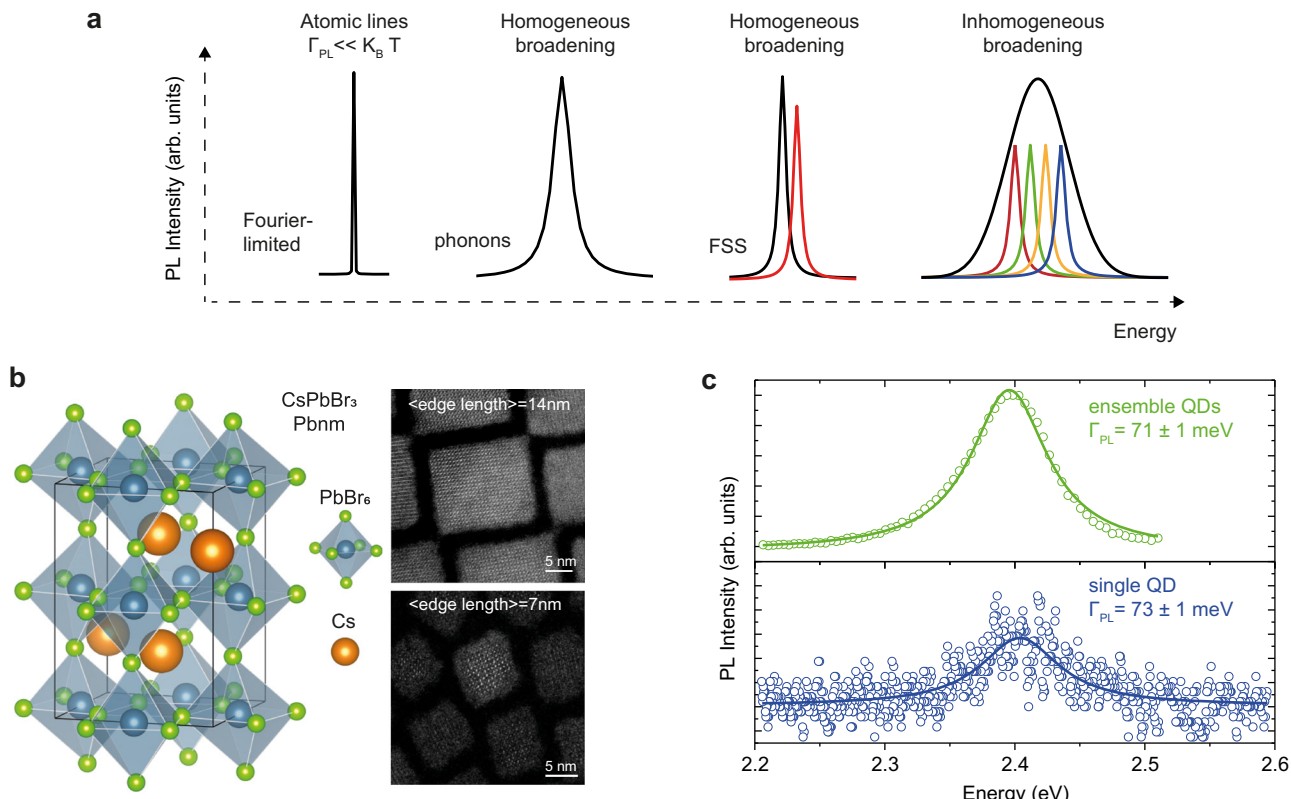

**Fig. 1 Emission line-broadening in nanomaterials: the case of perovskite compounds. a** A sketch of different emission line-broadening mechanisms ($\Gamma_{PL}$ is the linewidth, $k_B T$ the thermal energy, and FSS the fine structure splitting). **b** Schematic of the CsPbBr₃ crystal structure with two HAADF-STEM images, representing the two studied QD batches with mean edge lengths of 7 and 14 nm, respectively. **c** Room-temperature emission spectra at the ensemble and single particle levels. Both PL spectra feature similar emission line-broadening. Solid lines are best fits of a Lorentzian function to the experimental data (colored open circles).

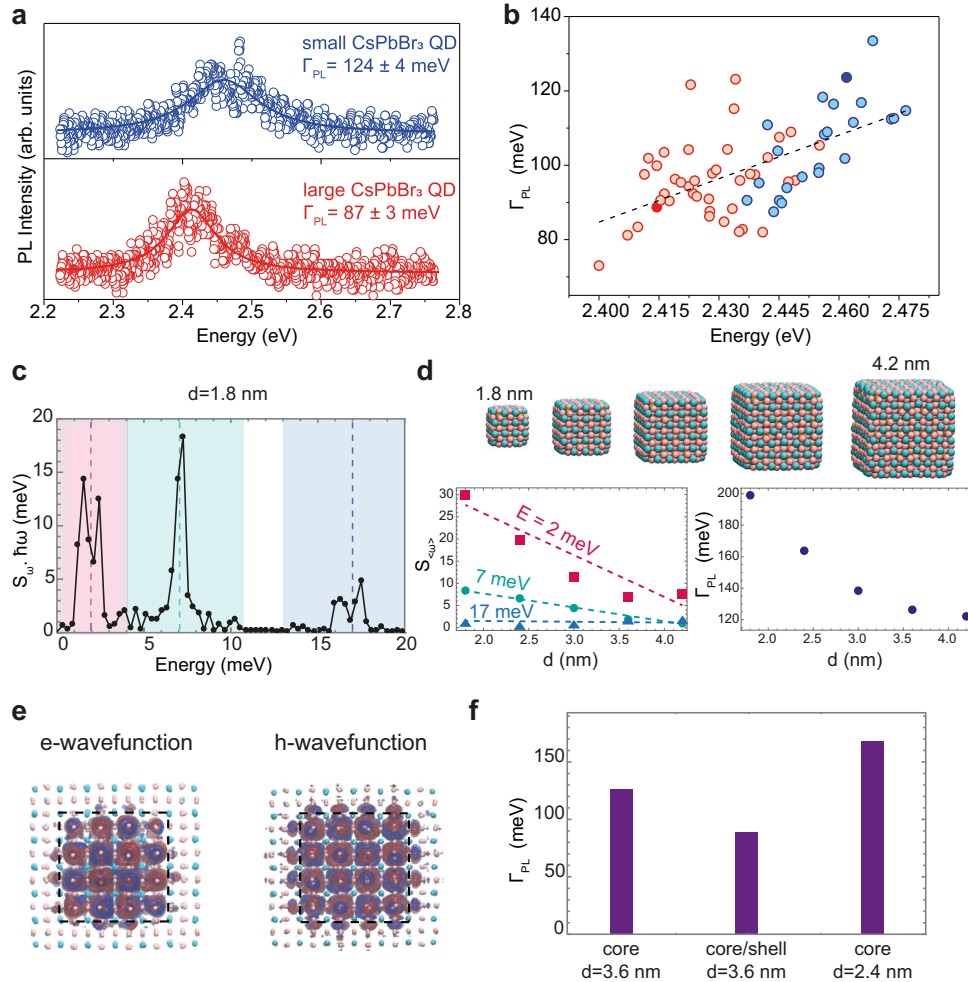

**Fig. 2 Size-dependent emission line-broadening in perovskite QDs and its origin. a** Two representative PL spectra of single QDs with different sizes. Solid lines are best fits of a Lorentzian function to the experimental data (colored open circles). **b** Statistics of emission line-broadening for various QDs: red and blue circles denote the emission broadening of QDs from two different batches. The circles filled in dark red and dark blue denote the emission linewidth values of the PL spectra in **a**. **c** The reorganization energy $S_\omega \cdot \hbar\omega$ as a function of the phonon energy $\hbar\omega$ ($S_\omega$ is the Huang–Rhys parameter for the lowest energy band-to-band transition). The different phonon modes are highlighted by shaded rectangles and dashed lines. The low-to-intermediate energy modes (2 and 7 meV) dominate the broadening. **d** Computationally explored QD size range from 1.8 to 4.2 nm (top plot). Computed size-dependent Huang–Rhys factor $S_{\langle\omega\rangle}$ (left plot) for the three phonon modes, where $\langle\omega\rangle$ indicates an effective coupling assuming a single mode in each of the three regions shown in **c** indicated by the dashed lines (see Supplementary Eq. (6)): while the high-energy mode (17 meV, blue triangles) has no dependence vs. QD size, strong variation is observed for the low-to-intermediate energy modes (red squares and green circles, respectively). Colored dashed lines are guide for the eye. The calculated emission broadening $\Gamma_{PL}$ (blue circles) vs. QD diameter is reported in the plot on the right. **e** Calculated electron and hole wavefunctions in a CsPbBr$_3$/CsCaBr$_3$ heterostructure, showing a clear type-I alignment. **f** Comparison of calculated line-broadening for core-only and core/ shell QDs: for QDs with an edge length of 3.6 nm, the emission in a core/shell QD (90 meV) is narrower than in a core-only QD (126 meV). The shell-induced PL narrowing is even more significant when compared to a core-only QD emitting at the same energy, i.e. a QD with an edge length of 2.4 nm exhibiting a PL broadening of 164 meV.

phonon modes at the QD surface. We show that mild modification of the QD surface leads to a type-I-like heterostructure, thereby reducing excitonic coupling to the surface phonon modes and hence the emission line-broadening. Record-narrow PL linewidth of 35 meV at room temperature is attained.

## Results and discussion

To understand the origin of emission line-broadening in perovskite QDs, we perform single QD spectroscopy in a home-built micro-PL setup. The main results of this work are obtained using two samples of CsPbBr$_3$ QDs having different mean edge lengths of 7 and 14 nm (Fig. 1b and Supplementary Fig. 1), which are prepared following a previously published synthetic protocol[10]. QDs are stabilized with long-chain zwitterionic ligands containing

ammonium and sulfonate functional groups[10]. They exhibit orthorhombic (*Pbnm*) perovskite crystal structure consisting of corner-sharing PbBr$_6$ octahedra with Cs ions filling the interoctahedral voids (Fig. 1b). High-resolution scanning transmission electron micrographs (Fig. 1b) confirm the high degree of size uniformity and the cuboidal QD shape. The ensemble PL linewidth of 71 meV for 14 nm QDs (Fig. 1c) is commensurate with those found in CsPbBr$_3$ QDs of the same size synthesized with alternative methods[9,11,25]. To confirm that the PL linewidth reflects homogenous, rather than inhomogeneous broadening, we compare the ensemble and single QD linewidths. The latter has been obtained via single QD spectroscopy on a film consisting of sparsely distributed single QDs in polystyrene matrix[26] prepared via spincoating on a glass coverslip (see the "Methods" section for more details). Despite having a size distribution of around 15%

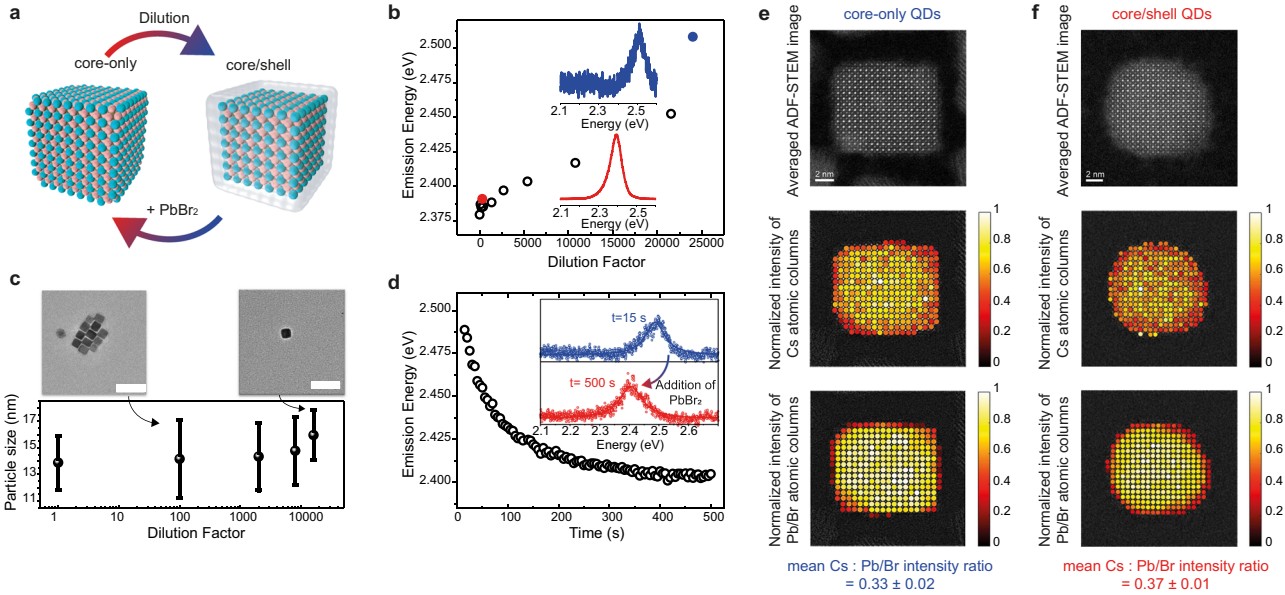

**Fig. 3 Dilution-induced surface modification of CsPbBr₃ QDs. a** A sketch of the core/shell formation during dilution and its reversibility. **b** Peak emission energy of QD dispersions upon progressive dilution (circle points). For a small (red solid marker) and a high dilution factor (blue solid marker), respectively, the insets show the corresponding PL spectra. **c** TEM analysis of particle size distribution at different dilution steps. Error bars represent the standard deviation of the size distribution obtained by measuring particle sizes for ca. hundred different QDs from the same batch. Two representative TEM images are also displayed (scale bar = 50 nm). **d** Addition of PbBr₂ to the QD solution redshifts, within several tens of seconds, the emission energy towards the original values before dilution (black open circles). Inset shows PL spectra at the beginning and after 500 s. **e, f** Typical ADF-STEM images of a core-only QD (**e**) and a core/shell QD (**f**) obtained through aligning and averaging their respective image series consisting of 10 frames (1024 × 1024, 1 μs dwell time); fit of the intensities of the Cs (middle plots) and Pb/Br atomic columns (lower plots) plotted at the fitted coordinates. For the sake of clarity and comparability, the fitted intensities of each QD are normalized to the maximum atomic column intensity. The analysis reveals that the core/shell QDs (see SI for additional examples) feature an amorphous, 1-nm-thick shell and a more spherical shape than original QDs. Additionally, the mean Cs:Pb/Br intensity ratio of the crystalline region of all studied QDs increases from 0.33 ± 0.02 in core-only QDs to 0.37 ± 0.01 in core/shell QDs, due to the leaching of Pb and Br ions upon dilution (errors denote the standard deviation).

(see Supplementary Fig. 1), these QDs exhibit an ensemble PL linewidth similar to that obtained by single QD spectroscopy (FWHM = 73 meV, Fig. 1c).

Representative PL spectra of a single QD from each of the two QD batches, along with broader sampling of QDs with different edge lengths are shown in Fig. 2a and b, respectively. A linear increase in emission line-broadening for higher emission peak energies (i.e. smaller sizes) is in agreement with an earlier study using solution-phase photon-correlation Fourier spectroscopy[25], a technique that allows the investigation of single QD optical properties in their native colloidal environment, with minimally invasive sample preparation. A similar trend is also observed when comparing ensemble PL spectra of QDs with different edge lengths (see Supplementary Fig. 2 and ref. [27]), and for much smaller QDs (e.g. QD edge length smaller than 6 nm), obtained by a size-selective precipitation method (see Supplementary Fig. 3), in line with reported values[28].

To gain insights into the origin of the size-dependent emission line-broadening, we perform AIMD simulations[29], from which we can directly compute the phonon density of states[30] and estimate the electron–phonon coupling strengths within the harmonic approximation. We have here specifically favored the computation of realistically sized QDs at a lower level of theory over the computation of unrealistically small QDs at a higher level of theory (see Supplementary Information and the "Methods" section for further details).

The calculated phonon density of states (Supplementary Fig. 4) shows agreement with the experimentally measured density of states from inelastic X-ray scattering experiments[31]. The electron–phonon coupling strength, i.e. the frequency-dependent Huang–Rhys factor $S_\omega$, for a transition from the valence-band maximum (VBM) to the conduction-band minimum (CBM) can be computed from the adiabatic electronic structure (see SI for more details),

$$S_\omega = \left| \mathscr{F}\{E_{CBM}(t) - E_{VBM}(t)\} \right|^2 / (4\hbar\omega k_B T) \quad (1)$$

where $\omega$ is the phonon frequency, $E_{CBM}(t)$ and $E_{VBM}(t)$ the energy of CBM and VBM, respectively, $\hbar$ Planck's constant over $2\pi$, $k_B$ the Boltzmann constant, $T$ the temperature, and $|\mathscr{F}\{x\}|^2$ corresponds to the modulus squared of the Fourier transform of $x$. The computed electron–phonon coupling strengths as a function of phonon energy (Fig. 2c) demonstrate that the lowest energy transition responsible for the PL couples to both low- (~2 meV) and intermediate-energy (~7 meV) phonons and to a lesser extent to the high-energy LO-phonon mode (~17 meV). We iterate the calculations for QDs with different edge lengths (Fig. 2d) and find that while the LO-phonon coupling strength demonstrates no clear size-dependence, the coupling to the low- and intermediate-energy phonons decreases with increasing size. The room-temperature FWHM linewidth ($\Gamma$) due to homogeneous broadening can be calculated from the electron–phonon coupling strength computed at 10 K and assuming the harmonic approximation is still valid at higher $T$ (see Supplementary Information for further details):

$$\Gamma(T) = 2\sqrt{2\ln(2)}\sqrt{2\Lambda k_B T}, \quad \Lambda = \sum_\omega S_\omega \hbar\omega \quad (2)$$

The calculations predict that the linewidth decreases for larger QDs (Fig. 2d), in agreement with our own experimental results (Fig. 2b) as well as recent theoretical works on different QD formulations[32–35].

 

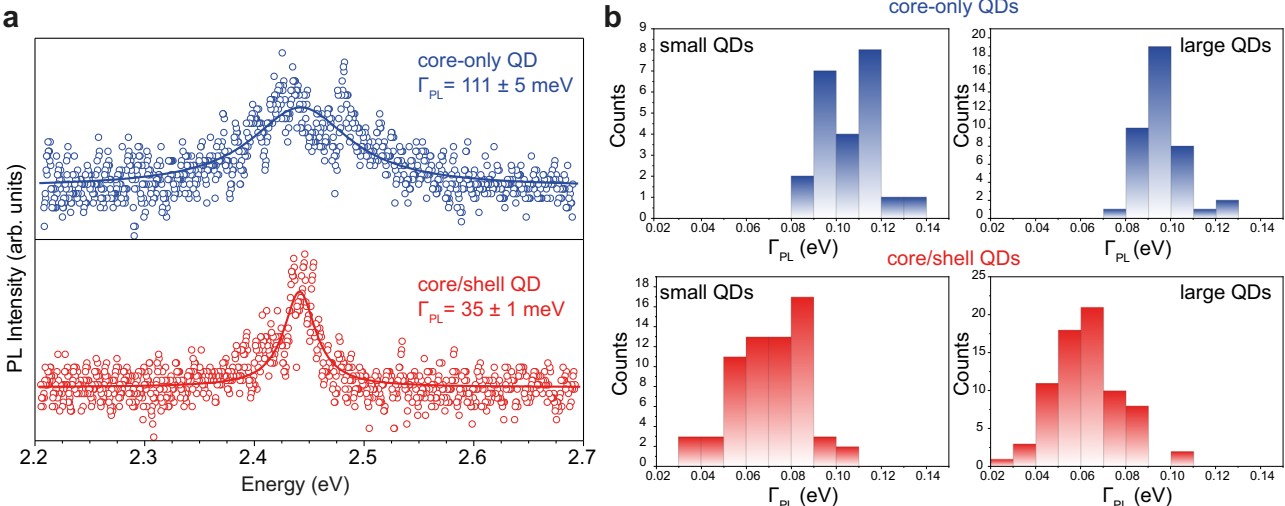

**Fig. 4 Experimental single-QD emission line-broadening in core-only and core/shell QD heterostructures. a** PL spectrum of a representative core-only QD (blue points) and of a core/shell QD (red points) exhibiting the record-low PL line broadening. Solid lines are best fit to a Lorentzian function. **b** Histogram of emission line-broadening for several core-only and core/shell QDs.

The strong size-dependence of the coupling to low- and intermediate-energy phonons could originate from two factors: (i) an increased coupling strength due to stronger quantum confinement of the carriers (relaxed momentum conservation rules), and/or (ii) increased coupling of the transition to phonon modes localized on the QD surface[31].

To distinguish between these two scenarios, we construct and simulate a type-I core/shell heterostructure, where both the electron and hole are confined in the QD core, reducing the overlap of the exciton with surface vibrations (Fig. 2e, Supplementary Fig. 5). We accomplish this by replacing the outer unit cells of the 3.6 nm QD with $CsCaBr_3$, chosen for its larger bandgap and lattice parameters which are nearly equal to that of $CsPbBr_3$, resulting in a strain-free heterostructure that has the same degree of quantum confinement as the 2.4 nm core-only QDs. Comparing the AIMD calculated thermal broadening for the 2.4 nm core-only, the 3.6 nm core-only, and the 3.6 nm core/shell QD (Fig. 2f), we find that the coupling to the low-energy surface phonon-modes and the linewidth progressively decreases (Supplementary Table 1). Hence, increased quantum confinement of charge carriers cannot explain the observed size-dependent PL broadening (Fig. 2d). On the contrary, calculations show that spatially confining carriers away from the surface is key to achieving narrow emission linewidth. The low-energy (2–7 meV) phonon modes have indeed been found by AIMD simulations to be highly sensitive to and enhanced at the QD surface (Supplementary Figs. 7–9), in analogy to other QD systems where undercoordinated atoms at QD surfaces give rise to surface-localized vibrations which couple to both inter- and intra-band transitions[31,36,37]. These calculations outline the possibility to achieve $CsPbBr_3$ QDs with narrow emission linewidths by developing a type-I core/shell heterostructure.

Unfortunately, a synthetic method for the epitaxial growth of a shell on perovskite QDs has to date remained elusive, a result of the structurally soft and chemically labile nature of metal halide perovskites. Nevertheless, we utilize the surface lability to induce a desired type-I alignment by gentle solution-phase treatments of these QDs. In particular, we build on the recent observations of a controlled removal of $Pb^{2+}$ and $Br^-$ ions from the $CsPbBr_3$ QD outer shell in response to the excess ligands present in the solution[38]. Deep progression of such etching (removal of $PbBr_2$)

can convert these QDs into $Cs_4PbBr_6$ or even CsBr QDs[38]. We sought the mildest possible etching that would address only the outmost surface layer of QDs, turning it into a wider-bandgap $CsPb_{1-x}Br_{3-2x}$ $(0 < x \leq 1)$ shell (schematic in Fig. 3a). We hypothesized that the very small but finite solubility of $PbBr_2$ in moist apolar solvents might be a sufficient driving force for $PbBr_2$ leaching from the QD surface, especially in highly dilute QD dispersions due to the mass-action effect (residual water-to-QD molar ratio). Such surface modification is then expected to cause a blueshift in the ensemble and single QD PL spectra. Hence, we have systematically explored the effect of dilution with various apolar solvents with different residual water contents on the resulting PL spectra (Fig. 3b, Supplementary Fig. 10). In particular, the emission energy of different QD toluene solutions that have been progressively diluted down to ca. 0.1 µg/mL (water content of ca. 200 ppm, Supplementary Fig. 10) undergoes a substantial blueshift by ca. 120 meV (Fig. 3b, Supplementary Fig. 10). A complementary analysis by transmission electron microscopy (TEM, Fig. 3c) reveals no substantial change in the average QD edge length, in line with the expected QD surface chemical modification. The magnitude of the energy blueshift correlates well with the trace water content in the solvent (Supplementary Fig. 10). The surface modification is reversible: addition of $PbBr_2$ to the diluted solution redshifts the emission energy towards the original emission value (Fig. 3d). Consequently, an effective way to suppress QD surface modification is to use ultra-dry solvent, a strategy we have employed for the results presented in Fig. 2a, b (see the "Methods" section for more details). Surface modifications at the single particle level were probed by annular dark field scanning transmission electron microscopy (ADF-STEM, Fig. 3e, f, Supplementary Figs. 11 and 12). Figure 3e, f show, for a core-only and a core/shell QD, respectively, from top to bottom, the averaged ADF-STEM images of the QDs, the corresponding intensity maps of the Cs and the Pb/Br atomic columns and the Cs: Pb/Br intensity ratio. Note that the Pb and Br atoms are in the same column and hence they appear as single bright spot in the image. Importantly, as will be deduced in the following, the ADF-STEM images reveal that the core/shell QDs are terminated by a lead- and bromide-deficient crystalline inner shell and an amorphous outer shell. First, from Fig. 3e, f one can readily infer the presence of an amorphous shell

of ~1 nm thickness for the core/shell QDs, and the absence of such an amorphous shell in the original QDs. Second, regarding the crystalline part of the QD, the mean Cs: Pb/Br intensity ratio of core-only QDs (see Supplementary Figs. 11 and 12 for additional examples) is found to be $0.33 \pm 0.02$, while for the core/shell QDs it increases up to $0.37 \pm 0.01$. ADF-STEM simulations (Supplementary Fig. 13) rule out that the detected difference in Cs: Pb/Br intensity ratio is a consequence of thickness induced dynamic diffraction effects due to geometry differences between the QDs from the original and the diluted dispersions. Instead, these simulations suggest that the experimental evidence can be explained by surface-transformed QDs with an inner crystalline shell depleted of lead and bromide as compared to the original $CsPbBr_3$ QDs. Hence, the combination on an amorphous outer shell and a crystalline inner shell is the structural basis for effectively confining the electronic wavefunctions in the inner part of the QD. The mean lattice spacing of the inner $CsPbBr_3$ core remains unchanged after dilution-induced surface modifications (Supplementary Fig. 14), attesting the absence of a built-in strain in these core/shell QDs.

While surface lability is generally seen as a drawback, it can be now used for imparting different degrees of surface rigidity and testing the exciton–surface–phonon coupling strength. Specifically, Fig. 4a compares single QD emission from a core-only QD and core/shell QD emitting at 2.44 eV. The emission line-broadening reduces from 110 meV for the pristine QD to 35 meV for the surface-modified QD. Statistics on hundreds of QDs shows a decrease of the emission linewidth for core/shell QDs to a mean value of ~60–70 meV, approaching the best reported values of colloidal II–VI QDs[22,39]. This is in good agreement with AIMD simulations, which predict suppressed coupling to low-energy phonon modes in type-I heterostructures (Fig. 2f).

Narrow emission linewidth is not the only desired attribute of a single-photon source. We also find that core/shell QDs display a stronger anti-bunching behavior, because of the stronger quantum confinement and minimal spectral fluctuations compared to the core-only QDs, establishing core/shell QDs as more suited single-photon sources (Supplementary Figs. 15 and 16). In addition, surface modifications and the core/shell formation do not significantly affect the brightness/quantum efficiency of single QDs (Supplementary Fig. 17) and the blinking dynamics (Supplementary Fig. 18) which still require further efforts for a complete suppression.

In summary, we have demonstrated that emission line-broadening in perovskite QDs at room temperature is dominated by coupling to low-energy phonon modes located predominantly at the surface of QDs. Just as lability along with dynamic surface ligand binding enables efficient ion-exchange transformations of perovskite QDs[40,41] and controlled post-synthetic surface treatments[11,42] lead to improved QD stability and higher PL QY, here we show that minimal surface modifications can have a profound impact on the PL emission linewidth and single-photon purity. Using these insights, further rational engineering of the QD surfaces could enable perovskite-based quantum emitters with sub-thermal, room-temperature emission linewidths, pivotal for light emitting devices and quantum technologies.

## Methods
### Synthesis of $CsPbBr_3$ QDs from cesium and lead oleates and $TOPBr_2$ or OAmBr

*Synthesis of cesium oleate.* $Cs_2CO_3$ (1.628 g, 5 mmol, containing 10 mmol Cs, i.e. 1 eq.) and oleic acid (5 mL, 16 mmol, 1.6 eq.) were evacuated in a three-neck flask along with 20 mL of ODE at room temperature until the gas evolution stops and then further evacuated at 25–120 °C for 1 h. This yields a 0.4 M solution of Cs-oleate in ODE. The solution turns solid when cooled to room temperature and was stored under argon and re-heated before use.

*Synthesis of lead(II) oleate.* Lead (II) acetate trihydrate (4.6066 g, 12 mmol, 1 eq.) and oleic acid (7.6 mL, 24 mmol, 2 eq.) were evacuated in a three-neck flask along with 16.4 mL of ODE at room temperature until the gas evolution stops and then further evacuated at 25–120 °C for 1 h. This yields a 0.5 M solution of Pb-oleate in ODE. The solution turns solid when cooled to room temperature and was stored under argon and re-heated before use.

*Synthesis of $TOP−Br_2$.* $TOPBr_2$ solution was prepared by mixing TOP (6 mL, 13 mmol, 1 eq.) with $Br_2$ (0.6 mL, 11.5 mmol, 0.88 eq.). The reaction between the two components is exothermic and requires vigorous stirring due to the product being highly viscous (white, almost solid). In order to make use of it as a precursor for injection, it was dissolved in toluene (18.7 mL) to form a 0.46 M light-yellow stock solution. The reaction was carried out in a Schlenk flask under argon.

*Synthesis of OAmBr.* Oleylammonium bromide was synthesized from oleylamine (62.5 mL, 0.19 mol, 1 eq) and aqueous HBr (0.19 mol, 1 eq) in a 500-mL ethanol solution and purified by recrystallization from diethylether and ethanol.

*Synthesis of $CsPbBr_3$ QDs*

14-nm $CsPbBr_3$ QDs: C3-sulfobetaine (21.5 mg, 0.1 mmol, 0.5 eq.) was mixed with lead oleate (0.5 mL, 0.25 mmol, 1.6 eq.), Cs oleate (0.4 mL, 0.16 mmol, 1 eq.) and 5 mL ODE in a 25 mL three-necked flask. The flask was equipped with a thermocouple sensor and a reflux condenser. The mixture was rapidly heated to 130 °C under vacuum and then to 180 °C under $N_2$ atmosphere. At 180 °C, TOP-$Br_2$ solution (0.5 mL, 0.46 mmol, 2.3 eq.) was swiftly injected using a 2-mL syringe. The reaction was stopped by cooling with an ice bath after 10–15 s. The cooling was removed at 25 °C.

The QDs were precipitated from the crude solution by centrifugation at 20,133×g (12,100 rpm) for 10 min. Subsequently, they were dispersed in 5 mL of toluene and precipitated with 10 mL of ethyl acetate and centrifuged at 20,133×g for 1 min, followed by redispersion in 5 mL of toluene. Insolubles were removed by centrifugation at 20,133×g for 10 min and the supernatant was filtered with a 0.45 μm syringe filter and stored in toluene. The entire purification process was conducted in air and analytic grade solvents with equilibrated water content were used (toluene ca. 200 ppm water, ethyl acetate ≤500 ppm).

7-nm $CsPbBr_3$ QDs: C3-sulfobetaine (461.1 mg, 2 mmol, 1.3 eq.) was mixed with lead oleate (5 mL, 2.5 mmol, 1.6 eq.), Cs oleate (4 mL, 1.6 mmol, 1 eq.) and 50 mL ODE in a 100 mL three-necked flask. The flask was equipped with a thermocouple sensor and a reflux condenser. The mixture was rapidly heated to 130 °C under vacuum, as soon as 130 °C were reached, the atmosphere was changed to nitrogen. In parallel, OAmBr (3.371 g) was mixed with 10 mL toluene and heated until dissolved. At 130 °C, warm OAmBr solution (≈10 mL) was swiftly injected into the Cs–Pb precursor solution using a 10 mL syringe. The reaction was stopped by cooling with an ice bath after 10–15 s. The cooling was removed at 25 °C.

To the crude solution (69 mL), 140 mL of ethyl acetate were added and the mixture was centrifuged at 20,133×g for 10 min. The supernatant was discarded and the precipitate was redispersed in 20 mL of toluene and precipitated with 40 mL of ethyl acetate and centrifuged at 20,133×g for 1 min, followed by redispersion in 10 mL of toluene. Insolubles, if any, were removed by centrifugation at 20,133×g for 10 min and the supernatant was filtered with a 0.45 μm syringe filter and stored in toluene. The entire purification process was conducted in air and analytic grade solvents with equilibrated water content were used (toluene ca. 200 ppm water, ethyl acetate ≤500 ppm).

*Sample preparation for single QD measurements.* An original QD solution (14-nm $CsPbBr_3$) was diluted 200–300 times (depending on initial concentration) with toluene (anhydrous). In the case of 7-nm $CsPbBr_3$ QDs, the use of anhydrous toluene alone was insufficient, due to the more labile ligand shell in the OLAmBr/C3-sulfobetaine mixed ligand system. Therefore, a saturated solution of OLAmBr was prepared and diluted 100 times with anhydrous toluene. The so-received solution can prevent a facile desorption of OLAmBr from the QD surface[42] and hence was used instead of toluene. In a second dilution step, each of the upon-named solutions was further diluted by adding 20 μL of it to a mixture of a 5 wt% polystyrene solution in anhydrous toluene (200 μL) and pure anhydrous toluene (800 μL). Polystyrene encapsulation was essential to prevent photo-degradation[26].

The so-received solutions were spin-coated onto glass (coverslip; thickness 170 ± 5 μm; diameter 25 mm; from Thorlabs) at 3000 rpm for 1 min using a Chemat Technology Spin-Coater (KW-4A).

*QD surface modifications.* Trace amounts of water can partially transform $CsPbBr_3$ QD surface, resulting in a core–shell heterostructure. The equilibrium water content of toluene under ambient conditions (ca. 200 ppm) is sufficient to cause this transformation in the upon-named dilution schemes. To achieve core–shell QDs, in both cases (7 and 14-nm $CsPbBr_3$ QDs), toluene with equilibrated water content (moist toluene) was used for the first dilution step (200–300 times). In a second dilution step, the QD solution was further diluted by adding 20 μL of it to a mixture of a 5 wt% polystyrene solution in moist toluene (200 μL) and moist toluene (800 μL).

The so received solutions were spin-coated onto glass (coverslip; thickness 170 ± 5 μm; diameter 25 mm; from Thorlabs) at 3000 rpm for 1 min using a Chemat Technology Spin-Coater (KW-4A).

Samples for transmission electron microscopy were prepared by diluting $CsPbBr_3$ QDs (1 mg/mL) 12,000 times with water saturated toluene (300 ppm water). Such dilution shifted the emission peak from initially 514 nm to below 480 nm in the diluted sample (see Supplementary Fig. 2b). The so received solution was drop-cast onto a Formwar/carbon covered copper TEM grid (mesh 200) that was placed on a filter paper. To receive a useful QD density on the grid 10 μL drops were dropped 10 times, waiting for the grid to dry between additions. As a reference sample, the undiluted solution of QDs was used.

### Structural and optical characterization

*TEM analysis.* TEM images were collected with a Hitachi HT7700 operated at 100 kV and HAADF-STEM images with a Hitachi HD2700 at cryogenic temperatures. All TEM images were processed using Fiji[43].

*Scanning transmission electron microscopy characterization.* The $CsPbBr_3$ QDs were studied at room temperature by annular dark-field scanning transmission electron microscopy (ADF-STEM) using a probe-corrected FEI Titan Themis microscope operated at 300 kV and setting a probe convergence semi-angle of 18 mrad. In order to limit electron beam damage, low dose ADF-STEM imaging was performed at an electron probe current of <2 pA in combination with collecting semi-angles of 35–190 mrad for the annular dark-field detector. The ADF-STEM images were obtained through aligning and averaging image series consisting of 10 frames (1024 × 1024, 1 μs dwell time) by means of the SmartAlign software[44]. For each frame of the series, the electron dose was 96 electrons/Å$^2$ and the dose rate at the given magnification was 96 electrons/Å$^2$ s. In addition to the noise-reduction by the summation of the individual frames, the series of images also warrant that the particles do not damage while being exposed to the low-dose electron beam.

*PL spectroscopy.* PL spectra were obtained using a Fluorolog iHR 320 Horiba Jobin Yvon spectrofluorimeter equipped with a PMT detector.

*Single QD spectroscopy.* For single QD spectroscopy, a home-built optical microscope was used. The sample was excited by a pulsed laser (<70 W/cm$^2$, 10 MHz laser emission at 405 nm), which is focused ($1/e^2 = 1$ μm) by an oil immersion objective (NA = 1.3). The emitted PL is collected by the same objective and sent to a monochromator coupled to an EMCCD camera. Alternately, the PL was sent to an HBT setup equipped with two APDs (time resolution = 250 ps) for second-order correlation measurements.

## Data availability
The data that support the findings of this study are available from the corresponding authors upon reasonable request.

## Code availability
All code used in this study is available from the corresponding authors upon reasonable request.

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

## Acknowledgements

G.R. acknowledges B. Benin and Dr. M. Bodnarchuk for useful discussions. M.V.K. acknowledges financial support from the Swiss Innovation Agency (Innosuisse, grant 32908.1 IP-EE) and, in part, from the European Union through Horizon 2020 research and innovation program (grant agreement No. [819740], project SCALE-HALO). I.I. acknowledges The Netherlands Organization of Scientific Research (NWO) for financial support through the Innovational Research Incentive (Vidi) Scheme (Grant No. 723.013.002) and S.C.B. acknowledges NWO for financial support through the Innovational Research Incentives (Veni) Scheme (Grant No. 722.017.011). N.Y. and V.W. acknowledge the Swiss National Supercomputing Centre (CSCS; project ID s1003). Funding for N.Y. was provided by the Swiss National Science Foundation through the Quantum Sciences and Technology NCCR. The project was also partially supported by the Air Force Office of Scientific Research and the Office of Naval Research under award number FA8655-21-1-7013, by the by the European Union's Horizon 2020 program, through a FET Open research and innovation action under the grant agreement Grant No. 899141 (PoLLoC) and by the Swiss National Science Foundation (Grant No. 188404, "Novel inorganic light emitters: synthesis, spectroscopy and applications").

## Author contributions

M.K.-C., C.Z., and G.R. performed the single QD optical characterization. F.K. synthesized and characterized QDs. N.Y. and S.C.B. performed ab-initio calculations under the supervision of V.W. and I.I. M.D.R. and R.E. performed the annular dark-field scanning transmission electron microscopy experiments and interpreted the results. All authors contributed to the interpretation of the results. G.R. and M.V.K. supervised the project and wrote the manuscript with input from all authors.

## Competing interests

The authors declare no competing interests.
