## [Peer Review File · Nature Communications]

Title: Ultra-Narrow Room-Temperature Emission from Single CsPbBr₃ Perovskite Quantum DotsREVIEWER COMMENTS

Reviewer #1 (Remarks to the Author):

The manuscript by Raino et al. reports a combined experimental and computational study of the photoluminescence (PL) linewidth in a series of CsPbBr₃ nanoparticles. The authors characterize the coupling of electronic transitions (HOMO-LUMO) responsible for PL to the nanoparticles' phonons. They find that the low-frequency phonons (at about 2 and 7 meV) are the major ones. The corresponding Huang-Rys factors at these frequencies are found to be dependent on the nanoparticle size, whereas the modes at higher energies (around 17 meV) are size-independent. These relationships explain lowering the PL linewidth with the increase of nanoparticle size.

The lower-energy phonons responsible for PL transition are found to be localized on the surface of nanocrystals. Hence, the authors explore the idea of increasing the "rigidity" of the outer part of NCs. In computer simulation, this can be achieved via "artificial" freezing of the atomic motion in the outer shell of perovskite NCs. The practical way of doing this is via creating a core/shell heterostructure. The authors demonstrate that this approach does indeed work, which is the main point of the manuscript. Overall, the work is very-well prepared: all the essential experimental/computational details are provided, the conclusions are well supported by the data, the data are well presented and explained. This is a solid, carefully-conducted work that addresses an important question of NCs' spectroscopic properties engineering. The results are substantial and should be published eventually.

Although at the level presented, the work is self-consistent, some limitations of the computational approach are apparent. First, the authors use only a single-particle picture (HOMO-LUMO) to describe the PL linewidth. How important are the excitonic (static-correlation) effects for describing the PL linewidth? How much would the use of the proper excited states (e.g. at the TD-DFT level) affect the quantitative and especially qualitative conclusions of the work presented? Second, the PBE functional is used, which is known to underestimate the computed band gaps. The authors use the Fourier transforms of the band gap to compute the Huang-Rys factors. I wonder how much such an underestimation of the BG would affect the calculations, especially from the qualitative point of view. From the non-technical perspective, my other concern is that the results presented are quite predictable, and not that surprising. It is well-known that the PL linewidth is controlled, in short, by the magnitude of the excitation energy fluctuation. In turn, this fluctuation is controlled by the structural "rigidity" of the system, where the corresponding modes are localized (the surface in this case). Thus, the control of the PL linewidth via surface modification is not a totally new concept. Some quick search for related topics yields a number of computational [e.g. J. Chem. Phys. 2001, 154, 214502; ACS Nano 2009, 3, 2487; J. Phys. Chem. C 2015, 119, 27954; NanoLett.2016, 16, 289] and experimental [e.g. J. Phys. Chem. C 2020, 124, 22; Nanoscale, 2020,12, 13113; NanoLett.2016, 16, 289] studies that have discussed similar (to some extent) mechanisms of the PL control. I should note, however, that although the concept is not entirely new, the current work makes a great connection between the experimental measurements and the underlying theory. The authors make a good job presenting a coherent, self-contained explanation. Thus, I believe this work could still be of interest to the broad readership such as that of Nature Communications.

Reviewer #2 (Remarks to the Author):

The authors combine experiments and ab-initio molecular dynamics to trace the origin of linewidth broadening in perovskite QDs to low-energy phonon coupling. By altering the surface of the perovskite QDs, they spatially confine carriers away from the surface, resulting in linewidth narrowing. This yields a record linewidth value (35 meV), compared with other perovskite materials at room temperature. I recommend publication of this manuscript in Nat comm after the following items have been acted upon:

1. In the manuscript, the authors describe their key method as a surface treatment in the abstract and conclusion but have mentioned multiple times a core-shell structure in the body text. I can agree that some similarities exist between shelling and the surface alternation in this manuscript, but the current description can be confusing. The surface layers here are much thinner than a shell and less crystalline. It would be useful for the authors to find a more proper/consistent description of the surface layer.
2. Similar to question 1, in Fig. 2e, it seems that the QD is embedded in a CsCaBr₃ shell to simulate a core-shell structure. As the 'shell' is still located in the surface layers, I think that the surface of the QD would still play a key role in the properties of the perovskite QDs. Also, there could be some differences in the calculation of CsCaBr₃ crystals and amorphous-like CsPbBr_x surface layers. It would be useful for the author to show how the state of the surface (surface ligand, atoms, vacancies) is considered during the calculation.
3. In Fig. 3e&f, the shape of the QD is altered after the mild etching process and the Cs:Pb/Br ratio is changed. Could this be related to the exposure of more facets after etching, compared with the core-only? Is it possible to exclude the possibility that this could alter the Cs:Pb/Br ratio and have a considerable effect on the linewidth narrowing?
4. In the text, the authors state that the emission line-broadening is reduced from 110 meV to 35 meV. However, from Figure 4b, it seems like the 110 meV corresponds to the average value of pre-treated dots, while the 35 meV corresponds to the best value for post-treated dots. Would it be more consistent to keep the numbers in this claim either the best or the average?
5. Before treatment, the ensemble and single molecule linewidth were roughly equivalent. It appears that the treatment results in more inhomogeneity in the dots. Is this just from different degrees of etching?

Reviewer #3 (Remarks to the Author):

The main claim of this report is that surface modifications can be used to create ultra-narrow linewidth

emission CsPbBr₃, with evidence for a mechanistic model through surface phonon interactions. This is claimed on basis of a large single dot spectroscopy data set, a new surface modification method, and ab-initio MD calculations.

It should be clear up front that I have no expertise in MD simulations, and only have expertise in single dot spectroscopy as a prospective user of dots as single photon emitters, and not from a synthesis viewpoint. From my viewpoint, a main obstacle for the use of CsPbBr₃ dots has been that there has been no equivalent of the core/shell(/shell) geometry that has allowed the engineering of CdSe based dots to reach high quality far exceeding that of the originally reported simple (core-only) dots. Thus I find the overall main claim appealing, and believe it could have important impact as this may be route to far extend the utility of CsPbBr₃ for diverse research fields.

From the viewpoint of validity, since I have no basis in the theory, I can not go further then observing that the theory and the data tell a consistent mechanistic story that is clearly explained, and has a strong support from the data portfolio. At the same time, I believe that on the data side many questions remain unanswered, although the answers can be teased out of the data. These questions are my suggestions for revision:

1. A main observation I have is that all vertical axes and color scales that relate to intensity / count rates for literally all of the graphs are in arbitrary units (even lacking 0's in crucial graphs, such as spectra in Figure 1c, 2a, etc). Since in the quantitative single photon emitter research field, count rates and brightness are crucial indicators of emitter quality and efficiency, a mandatory starting point should in my view be to report intensities quantitatively everywhere. I understand that there will be some set up specific prefactors with an uncertainty, but this should never be an excuse to withhold quantitative numbers.
2. A main question is how the surface modification quantitatively impacts brightness / quantum efficiency, either on ensemble level, or otherwise as judged from count rate histograms.
3. A crucial aspect in the work is spectral diffusion, and the authors show graphs with 100 consecutive spectra at 1 second intervals (supplement, which is in itself a rich treasure trove of additional results). This integration time seems very slow to me, given the expected intermittent dynamics of such dots, and also the technological possibilities for measuring spectra (at least down to 10 ms should be feasible in a usual single molecule microscopy set up). A comment on the dynamics / preferably faster data would be highly interesting.
4. Relating to point 2 and 3: these types of dots are intermittent (blinking), and a variety of mechanisms have been invoked, some of which involve the surface. Given the rich single-dot data sets the authors have, one would expect a discussion of the role of the surface chemical modification method on intermittency.
5. Regarding the data selection in Figure 4, the experimental spectrum in Figure 4a, in red is certainly impressively narrow, and I have no reservations per se about it being shown as a best result. However, to call it "representative" seems a stretch: the histogram in the same figure suggests that only 2 out of over 50 dots were as narrow. This makes it a "top 10%" dot.
6. The sales pitch of the paper comes from the perspective of atomic quantum memories, and Figure 1

advertises an atomically narrow, Fourier limited line (micro-eV, logically not in range at room temperature). How do the authors see a viable route to a room temperature quantum technology? The outlook would benefit from more concrete ideas on this front.

Response to the Referee Reports for Rainò *et al.* (NCOMMS-21-32776)

We sincerely thank all the Reviewers for their detailed reading of the manuscript and their constructive comments. We followed their suggestions in close detail to improve the manuscript. Below, please find the revisions in response to the Reviewers' comments point-by-point, with the comments of the Reviewers highlighted in blue and italic, followed by our reply and action. Overall, we have added several references and two new figures to the SI.

We are confident that our implemented changes have improved the manuscript and addressed the main open points raised by the Reviewers.

Referee #1

The manuscript by Rainò et al. reports a combined experimental and computational study of the photoluminescence (PL) linewidth in a series of CsPbBr₃ nanoparticles. The authors characterize the coupling of electronic transitions (HOMO-LUMO) responsible for PL to the nanoparticles' phonons. They find that the low-frequency phonons (at about 2 and 7 meV) are the major ones. The corresponding Huang-Rhys factors at these frequencies are found to be dependent on the nanoparticle size, whereas the modes at higher energies (around 17 meV) are size-independent. These relationships explain lowering the PL linewidth with the increase of nanoparticle size.

The lower-energy phonons responsible for PL transition are found to be localized on the surface of nanocrystals. Hence, the authors explore the idea of increasing the "rigidity" of the outer part of NCs. In computer simulation, this can be achieved via "artificial" freezing of the atomic motion in the outer shell of perovskite NCs. The practical way of doing this is via creating a core/shell heterostructure. The authors demonstrate that this approach does indeed work, which is the main point of the manuscript.

Overall, the work is very-well prepared: all the essential experimental/computational details are provided, the conclusions are well supported by the data, the data are well presented and explained. This is a solid, carefully-conducted work that addresses an important question of NCs' spectroscopic properties engineering. The results are substantial and should be published eventually.

Our response: we thank the Reviewer for the high appreciation of our work.

Although at the level presented, the work is self-consistent, some limitations of the computational approach are apparent. First, the authors use only a single-particle picture (HOMO-LUMO) to describe the PL linewidth. How important are the excitonic (static-correlation) effects for describing the PL linewidth? How much would the use of the proper excited states (e.g. at the TD-DFT level) affect the quantitative and especially qualitative conclusions of the work presented?

Our response: we thank the Reviewer for pointing out a typical challenge faced when attempting a dynamical description of excited states in semiconductor nanocrystals: while many-body descriptions (to include excitonic effects, e.g. at the TD-DFT level of theory) are already available for small molecules, at present such calculations are still prohibitively expensive for experimentally studied semiconductor nanocrystals with typical sizes of several nanometers (and thus thousands

of atoms). However, a very recent work by the Akimov group [see B. Smith et al., J. Chem. Theory Comput. 17, 678–693, (2021)] has carefully compared the single-particle and many-body description for (rather small) CdSe and Si nanocrystals in the framework of nonadiabatic dynamics and concluded that, while many-body effects do affect the dynamics at higher-energy excited states, the single-particle picture remains a valid description for the bandgap transition (which is the one we studied in our work on PL broadening). Hence, both the recent literature (on the validity of the single-particle picture) and the strongly size-dependent PL broadening (found experimentally and computationally in our own work) justifies our approach of favoring a computation of realistically sized nanocrystals over a computationally expensive inclusion of a higher level of theory with eventually minor accuracy improvement. In addition, the very good agreement between the experimental values and the computed ones, strongly support the claim that the level of theory implemented in our simulations is adequate to capture the experimental observations.

Our action: we have included a short paragraph to justify the level of theory employed in our simulations with the beforementioned reference.

Second, the PBE functional is used, which is known to underestimate the computed band gaps. The authors use the Fourier transforms of the band gap to compute the Huang-Rys factors. I wonder how much such an underestimation of the BG would affect the calculations, especially from the qualitative point of view.

Our response: The computed electron-phonon (EP) coupling strengths are independent of the magnitude of the bandgap, assuming the underestimation stems from a uniform redshift of the unoccupied bands. Any such (DC) shift would only contribute to the $S_{ij}(\omega = 0)$ term, with zero effect on the reorganization energy $\lambda_{ij}(\omega = 0) = S_{ij}(\omega = 0) \cdot \hbar\omega = 0$, and hence zero contribution to any radiative/non-radiative transitions. The results will depend, however, on the electron/hole effective masses, which can be impacted by the functional. While we agree that it would be great to check our results against other functionals and levels of theory, it is unfortunately computationally prohibitive for the large size of our experimentally and computationally studied nanocrystals (comprised of thousands of atoms). Nevertheless, the excellent agreement between our computational and experimental PL broadening suggests that the computed EP coupling strengths are not affected significantly by the usage of the PBE functional and all remaining approximations employed in this work.

Our action: we have clarified in the main text the approximations used in the calculations, and the expected impacts on simulated PL line broadening.

From the non-technical perspective, my other concern is that the results presented are quite predictable, and not that surprising. It is well-known that the PL linewidth is controlled, in short, by the magnitude of the excitation energy fluctuation. In turn, this fluctuation is controlled by the structural “rigidity” of the system, where the corresponding modes are localized (the surface in this case). Thus, the control of the PL linewidth via surface modification is not a totally new concept. Some quick search for related topics yields a number of computational [e.g. J. Chem. Phys. 2001, 154, 214502; ACS Nano 2009, 3, 2487; J. Phys. Chem. C 2015, 119, 27954; NanoLett.2016, 16, 289] and experimental [e.g. J. Phys. Chem. C 2020, 124, 22; Nanoscale, 2020,12, 13113; NanoLett.2016, 16, 289] studies that have discussed similar (to some extent) mechanisms of the PL control. I should note, however, that although the concept is not entirely new, the current work makes a great connection between the experimental measurements and the underlying theory. The authors make a good job presenting a coherent, self-contained explanation.

Thus, I believe this work could still be of interest to the broad readership such as that of Nature Communications.

Our response: we thank the Reviewer for this comment. Indeed, some insights of our work concerning the critical role of surface properties in relation to PL line broadening have been already introduced in literature, for other QD formulations. We have extended our reference list to account for some of the suggested works. However, a systematic study of size-dependent and composition-dependent (core vs. core/shell heterostructures) PL line broadening, coherently supported by AIMD simulations, is still missing especially for perovskite nanocrystals, a new member in the family of quantum dots featuring very soft crystal structure and relatively high exciton-phonon coupling. We firmly believe that the contribution of low-energy surface phonon modes, particularly strong in perovskite nanocrystals, was largely ignored in relation to the emission line broadening. No direct attempts to suppress their detrimental effects were undertaken up to now. Controlling line-emission broadening is undoubtedly pivotal for quantum dot LEDs, because the narrow PL linewidth is a key differentiator, which enabled QDs to enter and conquer the market of TV displays. On a longer term, narrow emission linewidth will be crucial for enabling quantum light sources, especially important when single photon emitters have to be interfaced with quantum memories. Alternatively, such efficient room temperature quantum light sources could be employed in single photon LEDs, whereas ultra-narrow and spectrally tunable emission could allow single photon multiplexing schemes [Nature Materials 17, 394–405 (2018)], significantly boosting the transfer rates in quantum communication.

Referee #2

The authors combine experiments and ab-initio molecular dynamics to trace the origin of linewidth broadening in perovskite QDs to low-energy phonon coupling. By altering the surface of the perovskite QDs, they spatially confine carriers away from the surface, resulting in linewidth narrowing. This yields a record linewidth value (35 meV), compared with other perovskite materials at room temperature. I recommend publication of this manuscript in Nat comm after the following items have been acted upon:

Our response: we thank the Reviewer for the positive assessment of our work. We have followed their recommendation in detail to improve the manuscript and answer the remaining open points.

1. In the manuscript, the authors describe their key method as a surface treatment in the abstract and conclusion but have mentioned multiple times a core-shell structure in the body text. I can agree that some similarities exist between shelling and the surface alternation in this manuscript, but the current description can be confusing. The surface layers here are much thinner than a shell and less crystalline. It would be useful for the authors to find a more proper/consistent description of the surface layer.

Our response: we agree that the shell in our perovskite QD systems, generated via surface modifications rather than a direct growth of a second semiconductor compound, is slightly different to what has so far been reported in other, more conventional heterostructures. While our shell is relatively thin and on the order of few nanometers only (similar to ZnS shells in CdSe/ZnS QDs which can be, likewise, only a few monolayers thick), the presence of the shell induced a significant PL blue shift with accelerated lifetimes, typical of type-I core/shell heterostructures. More

importantly, the shell can decouple the excitons from surface localized phonon modes, thus narrowing PL emission.

Our action: we have clarified in the main text the composition of the shell in order to avoid confusion given that the same terminology is largely employed to identify two different semiconductors in more conventional QD heterostructures.

2. Similar to question 1, in Fig. 2e, it seems that the QD is embedded in a CsCaBr₃ shell to simulate a core-shell structure. As the 'shell' is still located in the surface layers, I think that the surface of the QD would still play a key role in the properties of the perovskite QDs. Also, there could be some differences in the calculation of CsCaBr₃ crystals and amorphous-like CsPbBr_x surface layers. It would be useful for the author to show how the state of the surface (surface ligand, atoms, vacancies) is considered during the calculation.

Our response: as mentioned by the Referee, the type-I CsPbBr₃/CsCaBr₃ core/shell QD shown in Figure 2e was employed to computationally rationalize how narrow PL emission can be achieved in CsPbBr₃ QDs: both electron and hole wavefunction should be confined to QD regions away from the surface (see Fig 2e). Based on the computationally suggested emission-narrowing strategy, we have devised an experimental surface treatment which mimics the idealized type-I core/shell structure from the calculations and hereby successfully achieve record-narrow PL in perovskite QDs. As elaborated upon in the main text and SI, while the exact shell composition realized in the experiments is more complicated than shown in Figure 2e (and likely comprised of a PbBr₂-deficient Cs_xPb_yBr_z shell), the concept of wavefunction localization away from the QD surface is still applicable for such a shell. Attempts to simulate a shell of amorphous CsBr on top of CsPbBr₃ are difficult given the plethora of possible geometrical implementations of such an amorphous shell and the sensitivity of the electronic structure to each geometric arrangement. Therefore, simulations of amorphous shells are currently out of our computational reach and will be subject of future works.

Finally, while we agree with the Reviewer that, in general, synthetic control over ligands, atoms, and vacancies at the QD surface is typically pivotal for controlling PL properties in traditional perovskite QDs, the case is different in core/shell QDs: here, the wavefunction delocalization only over the core (and not shell) means that the wavefunctions (and thus PL broadening) have lost their sensitivity to processes at the QD (shell) surface. Hence, we believe that core/shell formation, rather than control over the QD surface (or even the interior part of the shell), is the crucial parameter which allowed us to achieve record-narrow PL in perovskite QDs.

3. In Fig. 3e&f, the shape of the QD is altered after the mild etching process and the Cs:Pb/Br ratio is changed. Could this be related to the exposure of more facets after etching, compared with the core-only? Is it possible to exclude the possibility that this could alter the Cs:Pb/Br ratio and have a considerable effect on the linewidth narrowing?

Our response: we thank the Reviewer for this comment. Indeed, as shown in Fig. 3e and 3f, as well as Figs. S11 and S12, the dilution-induced surface modifications alter the shape of the QD, transforming initially cuboidal QDs into spheroidal QDs (the concept of facets is here less applicable). Therefore, we choose to discuss dilution-induced modifications of the Cs:Pb/Br stoichiometry in terms of merely geometry-related effects (*i.e.* sphere vs. cube) or intrinsic Pb/Br deficiencies occurring at the QD surface. As demonstrated in the Supporting Information, geometry/shape effects cannot fully explain the differences in the observed Cs:Pb/Br stoichiometry for core-only and core/shell QDs, which is why we deduce that the outer (shell) QD layer in

core/shell QDs must be Pb/Br deficient, which – together with the amorphous layer – acts to confine the exciton away from the QD surface, hereby achieving narrow-band PL.

4. In the text, the authors state that the emission line-broadening is reduced from 110 meV to 35 meV. However, from Figure 4b, it seems like the 110 meV corresponds to the average value of pre-treated dots, while the 35 meV corresponds to the best value for post-treated dots. Would it be more consistent to keep the numbers in this claim either the best or the average?

Our response: we thank the Reviewer for this comment and for highlighting this inconsistency.

Our action: we have modified the main text and used the “record-low” or “average” adjective when referring to the experimental PL line broadening.

5. Before treatment, the ensemble and single molecule linewidth were roughly equivalent. It appears that the treatment results in more inhomogeneity in the dots. Is this just from different degrees of etching?

Our response: we thank the Reviewer for this comment. Indeed, each individual QD experiences a slightly different transformation of the outer layers, thus increasing the structural inhomogeneity of the ensemble. However, the PL linewidth of the core/shell QD ensemble is smaller than the PL linewidth of an equivalent core-only ensemble PL emitting at the same energy (see Figure S2, reported also here below). Such a net PL linewidth reduction at the ensemble level means that our surface transformation induces a reduction in PL line-broadening of individual QDs which is greater than the increase resulting from a larger inhomogeneous disorder at the ensemble level.

Figure S2| Ensemble PL spectra. a, Ensemble PL spectra for QDs with edge length 14 nm (red circle), 7 nm (green circle), 4.5 nm (blue circle). An increase in PL FWHM has been observed by reducing the QD size, from 73 meV up to 127 meV, in line with single QD spectroscopy results. **b,** ensemble PL spectrum before (red dot) and after (blue dots) dilution (x 12000). Solid lines are Lorentzian best fits of the experimental data. For core/shell QDs, the emission line broadening is much narrower than QDs emitting at the same energy (equivalent quantum confinement regime), in line with single QD spectroscopy results.

The main claim of this report is that surface modifications can be used to create ultra-narrow linewidth emission CsPbBr₃, with evidence for a mechanistic model through surface phonon interactions. This is claimed on basis of a large single dot spectroscopy data set, a new surface modification method, and ab-initio MD calculations.

It should be clear up front that I have no expertise in MD simulations, and only have expertise in single dot spectroscopy as a prospective user of dots as single photon emitters, and not from a synthesis viewpoint. From my viewpoint, a main obstacle for the use of CsPbBr₃ dots has been that there has been no equivalent of the core/shell(/shell) geometry that has allowed the engineering of CdSe based dots to reach high quality far exceeding that of the originally reported simple (core-only) dots. Thus I find the overall main claim appealing, and believe it could have important impact as this may be route to far extend the utility of CsPbBr₃ for diverse research fields. From the viewpoint of validity, since I have no basis in the theory, I cannot go further then observing that the theory and the data tell a consistent mechanistic story that is clearly explained, and has a strong support from the data portfolio. At the same time, I believe that on the data side many questions remain unanswered, although the answers can be teased out of the data.

Our response: we thank the Reviewer for the positive assessment of our work. We have followed their recommendation in detail to improve the manuscript and answer the remaining open points.

These questions are my suggestions for revision:

- 1. A main observation I have is that all vertical axes and color scales that relate to intensity / count rates for literally all of the graphs are in arbitrary units (even lacking 0's in crucial graphs, such as spectra in Figure 1c, 2a, etc). Since in the quantitative single photon emitter research field, count rates and brightness are crucial indicators of emitter quality and efficiency, a mandatory starting point should in my view be to report intensities quantitatively everywhere. I understand that there will be some set up specific prefactors with an uncertainty, but this should never be an excuse to withhold quantitative numbers.*

Our response: we thank the Reviewer for this comment and we agree that brightness is an important figure of merit for a single-photon emitter. In our work, we aimed to unveil the origin of the PL broadening and used low excitation power and short integration time (1 s) to avoid that spectral diffusion or dynamical PL blueshift could severely affect the PL broadening [Nano Lett. 19, 3648-3653 (2019)]. However, to account for the Reviewer request, we have performed additional measurements seeking to drive single perovskite QDs into the saturation regime. This should give an indication of the single-photon brightness of individual perovskite QDs. Given the fragile crystal structure of perovskite QDs, this is, however, not trivial. To achieve our goal, we have utilized a special encapsulation procedure to avoid photodegradation upon exposure to the laser light and moisture. In our measurements, to check for reversibility, we have recorded the power-dependent PL intensity by consecutively measuring both in forward direction (*i.e.* sweeping from low to high laser fluence) and reverse direction (*i.e.* sweeping from high to low laser fluence). This way, we ensured that any observed saturation levels are not due to photodegradation but rather due to the discrete nature of the excitation in a QD. While several QDs still suffered from photodegradation, manifested in a significant drop of PL counts already after the forward sweep,

the encapsulation did yield a sufficient number of QDs which were stable throughout the entire measurement cycle. Since such stable QDs exhibited similar power-dependent trends in both forward and reverse sweep, we posited that photodegradation played a marginal role, enabling a trustworthy comparison between the saturation levels in core-only and core/shell QDs.

The obtained results are summarized in Figure R1. For core-only QDs, typical saturation levels are on the order of $2.5 \cdot 10^5$ counts/s, similarly for core/shell QDs (ca. $2\text{-}2.5 \cdot 10^5$ counts/s). Therefore, qualitatively, the surface modifications and the core/shell formation do not significantly alter the brightness of the emitter. As mentioned before, the limited optical stability (despite encapsulation) and the resulting large dot-to-dot variation hinder a more quantitative analysis, which could be the subject of future work.

Nevertheless, the obtained count rates do provide an indication of the brightness of the perovskite QDs and, considering some variation in setup-specific detection efficiencies between this work and previously reported works, could be compared with other nanoscale emitters [e.g. Nature Photonics 11, 58–62 (2017)]. To further increase the brightness and detection efficiency, embedding single QDs into optical or plasmonic microcavities is an essential and consolidated strategy already explored for other single-photon sources.

Our action: we have included this new data in the SI, which, on a qualitative level, establishes that the surface modification and the core/shell formation do not significantly affect the brightness/quantum efficiency of single QDs.

Figure R1| Excitation power-dependent PL experiments. a, PL spectrum of a core-only QD emitting at 2.419 eV with a PL FWHM of 72 meV. **b,** two-dimensional false-color plots of PL spectra vs. excitation power, for two consecutive runs. **c,** spectrally-integrated PL count rate vs. excitation power, exhibiting a saturation level of about $2.5 \cdot 10^5$ counts/s. **d,** PL spectrum of a core/shell QD emitting at 2.517 eV with a PL FWHM of 71 meV. **e,** two-dimensional false-color plots of PL spectra vs. excitation power, for two consecutive runs. **f,** spectrally-integrated PL count rate vs. excitation power, exhibiting a saturation level of about $2\text{-}2.5 \cdot 10^5$ counts/s.

2. *A main question is how the surface modification quantitatively impacts brightness / quantum efficiency, either on ensemble level, or otherwise as judged from count rate histograms.*

Our response: as detailed in our previous response, we did not observe strong variations in terms of brightness / quantum efficiency upon surface modifications. Perovskite QDs do still experience strong instabilities upon photoexcitation, which render a more quantitative analysis not feasible. This could be the subject of future works.

3. *A crucial aspect in the work is spectral diffusion, and the authors show graphs with 100 consecutive spectra at 1 second intervals (supplement, which is in itself a rich treasure trove of additional results). This integration time seems very slow to me, given the expected intermittent dynamics of such dots, and also the technological possibilities for measuring spectra (at least down to 10 ms should be feasible in a usual single molecule microscopy set up). A comment on the dynamics / preferably faster data would be highly interesting.*

Our response: as reported in Figure S16 in the SI, spectral diffusion seems to not be strongly influenced by surface modifications, at least with our employed integration time of 1 s. Concerning the Reviewer's comment on the intermittency of the QDs: here, unlike for the slow spectral diffusion process, a shorter integration time is indeed needed. We gratefully have considered this suggestion by the Reviewer and will discuss it in the following point.

4. *Relating to point 2 and 3: these types of dots are intermittent (blinking), and a variety of mechanisms have been invoked, some of which involve the surface. Given the rich single-dot data sets the authors have, one would expect a discussion of the role of the surface chemical modification method on intermittency.*

Our response: we thank the Reviewer for this comment. We have performed additional experiments to elucidate whatever surface modifications alter blinking dynamics in single QD. The results obtained for core-only and core/shell QDs are reported in Figure R2. In both the core-only QD (Figure R2b) and the core-shell QD (Figure R2e), after performing the fluorescence lifetime intensity distribution (FLID) analysis [Nature 479, 203–207 (2011)], we have observed the typical A-blinking behavior, with low-intensity states exhibiting faster lifetimes. The ON/OFF ratio remains essentially unaltered. In conclusion, no dramatic change in blinking dynamics could be detected upon surface modifications.

Our action: we have included this new data set in the SI.

Figure R2 | Blinking dynamic in core-only (a-c) and core/shell QDs (d-f). **a**, PL spectrum of a core-only QD emitting at 2.4363 eV with a PL FWHM of 72.2 meV. **b**, The corresponding (fluorescence-lifetime intensity distribution) FLID colour plot. The two-dimensional histogram was constructed with a 0.1-ns lifetime binning and a 1 count intensity binning. The colour scale represents the frequency of occurrences of given intensity-lifetime pairs. **c**, *upper panel*: intensity histogram obtained with 1 ms binning; *lower panel*: blinking trace corresponding to the time window marked by a blue box in the upper panel. **d**, PL spectrum of a core/shell QD emitting at 2.497 eV with a PL FWHM of 70.3 meV. **e**, The corresponding core/shell QD FLID colour plot. The two-dimensional histogram was constructed with as in **b**. **f**, *upper panel*: intensity histogram obtained with 1 ms binning; *lower panel*: representative blinking trace corresponding to the time window marked by a blue box in the upper panel.

5. Regarding the data selection in Figure 4, the experimental spectrum in Figure 4a, in red is certainly impressively narrow, and I have no reservations per se about it being shown as a best result. However, to call it “representative” seems a stretch: the histogram in the same figure suggests that only 2 out of over 50 dots were as narrow. This makes it a “top 10%” dot.

Our response: we thank the Reviewer for this comment and for highlighting this inconsistency.

Our action: we have modified the main text and used the correct adjective (“record-low” or “average”) when referring to the experimental PL line broadening.

6. The sales pitch of the paper comes from the perspective of atomic quantum memories, and Figure 1 advertises an atomically narrow, Fourier limited line (micro-eV, logically not in range at room temperature). How do the authors see a viable route to a room temperature quantum technology? The outlook would benefit from more concrete ideas on this front.

Our response: controlling line-emission broadening is undoubtedly pivotal for quantum-dot LEDs, because the narrow PL linewidth is a key differentiator, which enabled QDs to enter and conquer the market of TV displays. Therefore, our work has an immediate impact on the development of LEDs and more in general down-converting display technologies employing perovskite QDs. On a longer term, narrow emission linewidth will be crucial for enabling quantum-light sources operating at room temperature, especially important when single-photon emitters have to be interfaced with quantum memories. In this scenario, which is still in its infancy, a more pronounced

reduction in PL linewidth is needed, as correctly pointed out by the Reviewer. A more realistic goal could be to obtain linewidths smaller than thermal energy, a result which has just been accomplished by precisely-engineered CdSe/Cd_xZn_{1-x}Se QDs [Nature Materials 18, 249–255 (2019)]. Alternatively, such efficient room-temperature quantum-light sources could be employed in single-photon LEDs, whereas ultra-narrow and spectrally tunable emission could allow single-photon multiplexing schemes [Nature Materials 17, 394–405 (2018)], significantly boosting the transfer rates.

Our action: we included a sentence in the main text to elucidate this point and give a more viable outlook on room-temperature quantum-light sources.

REVIEWERS' COMMENTS

Reviewer #1 (Remarks to the Author):

As I indicated in the original review, the work was almost ready for publication. The authors have adequately addressed my concerns and made corresponding changes to the revised manuscript. I'm happy to recommend the work to be published as-is. No further review is needed.

Reviewer #2 (Remarks to the Author):

The manuscript was already excellent, and suited for Nature Comms, in the previous round. Now the authors have answered my questions and acted on suggestions. I am satisfied that the work is now worthy of acceptance at Nature Comms.

Reviewer #3 (Remarks to the Author):

I thank the authors for the revisions which have addressed my suggested points for improvement. I can recommend the manuscript for publication.

P.S. Figure S3a does have a glaring typo in its labelling, which should be corrected.

Response to the Referee Reports for Rainò *et al.* (NCOMMS-21-32776A)

We sincerely thank all the Reviewers for their detailed reading of the manuscript and their constructive comments; they really helped to further improve the manuscript.

Referee #1

As I indicated in the original review, the work was almost ready for publication. The authors have adequately addressed my concerns and made corresponding changes to the revised manuscript. I'm happy to recommend the work to be published as-is. No further review is needed.

Our response: We thank the Reviewer for the high appreciation of our work.

Referee #2

The manuscript was already excellent, and suited for Nature Comms, in the previous round. Now the authors have answered my questions and acted on suggestions. I am satisfied that the work is now worthy of acceptance at Nature Comms.

Our response: We thank the Reviewer for the high appreciation of our work.

Referee #3

I thank the authors for the revisions which have addressed my suggested points for improvement. I can recommend the manuscript for publication.

P.S. Figure S3a does have a glaring typo in its labelling, which should be corrected.

Our response: We thank the Reviewer for the high appreciation of our work and for highlighting the typo in Figure S3a, which has now been corrected.